# Beneficial Effects of a Freeze-Dried Kale Bar on Type 2 Diabetes Patients: A Randomized, Double-Blinded, Placebo-Controlled Clinical Trial

**DOI:** 10.3390/nu16213641

**Published:** 2024-10-26

**Authors:** Per Bendix Jeppesen, Amanda Dorner, Yuan Yue, Nikolaj Poulsen, Sofie Korsgaard Andersen, Fie Breenfeldt Aalykke, Max Norman Tandrup Lambert

**Affiliations:** 1Department of Clinical Medicine, Aarhus University, Aarhus University Hospital, 8200 Aarhus N, Denmark; amanda.dorner@zvijezda.hr (A.D.); 201709604@post.au.dk (S.K.A.); fiejuul@hotmail.com (F.B.A.); mntl@clin.au.dk (M.N.T.L.); 2Department of Animal Science, Aarhus University, 8830 Tjele, Denmark; yuan.yue@anivet.au.dk

**Keywords:** kale, type 2 diabetes, vegetables, calorie intake, essential AA

## Abstract

Background/Objectives: Type 2 diabetes (T2D) is one of the most common global diseases, with an ever-growing need for prevention and treatment solutions. Kale (*Brassica oleracea* L. var. *acephala*) offers a good source of fiber, minerals, bioavailable calcium, unsaturated fatty acids, prebiotic carbohydrates, vitamins, health-promoting secondary plant metabolites, as well as higher amounts of proteins and essential amino acids compared to other vegetables. The objective of this study was to investigate whether daily intake of freeze-dried kale powder can provide health benefits for T2D patients vs. placebo. Methods: This study was designed as a 12-week, blinded, randomized, controlled trial. Thirty T2D patients were randomly assigned to either a placebo bar (control) or a kale bar (intervention). Participants in the intervention group were instructed to consume three bars/day, each containing 26.25 g of freeze-dried kale (corresponding to approx. 341 g fresh kale/day). At baseline and 12 weeks, all participants underwent an oral glucose tolerance test (OGTT), 24 h blood pressure measurements, DEXA scans, and fasted blood samples were taken. Results: A significant reduction in HbA1c, insulin resistance, body weight, and calorie intake was observed in the intervention group compared to control. Positive trends were detected in fasted blood glucose and LDL-cholesterol for those in the kale intervention group. No significant differences were found in total body fat mass and area under the curve glucose 240 min OGTT. Conclusions: Given the positive effects of high daily kale intake observed in this study, further research with a larger sample size is needed to better understand the health benefits of kale bars. This could potentially lead to new dietary recommendations for patients with T2D.

## 1. Introduction

Type 2 diabetes mellitus (T2D) is considered one of the most common diseases globally [1]. Diabetes is a major risk factor for heart attacks, stroke, blindness, kidney failure, and lower limb amputation. According to the International Diabetes Federation, 537 million adults were living with T2D in 2021 [1]. This number is projected to increase to 783 million adults by 2045. These proportions of affected population represent a great public health challenge all over the world. Unfortunately, some people with T2D develop serious complications such as cardiovascular disease, neuropathy, nephropathy, and retinopathy [2]. T2D is a complex disease characterized by hyperglycemia and disruptions to normal metabolism. These include reduced insulin sensitivity in various organs, elevated hepatic glucose production, decreased glucose uptake in skeletal muscles, increased lipolysis in adipose tissue, as well as impaired effects of incretin hormones, reduced insulin secretion from pancreatic beta cells, and increased glucagon secretion from pancreatic alpha cells [3]. Treatment of diabetes predominantly involves dietary interventions and increased physical activity with the goal to lower blood glucose and other known risk factors which may damage blood vessels. However, there is a growing need for practical, accessible, and cost-effective solutions for the prevention and treatment of diabetes [4].

Kale (Brassica oleracea L. var. acephala) is a green leafy vegetable from the Brassicaceae family. It is a good source of fiber and minerals (e.g., potassium) and retains high concentrations of bioavailable calcium (citrate or malate). Furthermore, kale also contains unsaturated fatty acids, prebiotic carbohydrates, various vitamins, and health-promoting secondary plant metabolites. It has also been reported that kale contains much higher amounts of protein and essential amino acids compared to other Brassica vegetables [5,6,7,8].

Freeze dried kale powder has already been successfully incorporated into some food matrices, such as cookies and muffins, to increase their functionality [9,10]. In this study, freeze-dried green kale powder was incorporated into a convenient bar and administered daily to T2D patients to investigate the potential health beneficial effects of 3 months of intake.

As T2D treatment mainly includes modifications to diet and exercise, all patients were recommended to include a variety of fresh fruits and vegetables in their meals every day [11,12,13]. According to a meta-analysis conducted by Wang et. al. (2016), higher intakes of fruits and green leafy vegetables, yellow vegetables, cruciferous veget½ables, or their fibers are associated with a lower risk of T2D [14]. Furthermore, some studies have shown correlations between the consumption of cruciferous vegetables and improvement of T2D [15,16,17]. According to Yossef et al. (2012), the intake of red cabbage demonstrated significant therapeutic effects in diabetic rats. The study found that the administration of dried red cabbage caused significant decreases in levels of blood glucose, triglycerides, total cholesterol, LDL and VLDL cholesterol, as well as significantly increased HDL cholesterol and improved liver and kidney function compared to control [18].

Positive effects in humans have also been demonstrated in clinical trials. Broccoli sprouts in the form of a powder were investigated by Bahadoran et al. (2012). The powder was administered as a supplementary treatment in T2D and demonstrated favorable effects on lipid profiles and OX-LDL/LDL cholesterol ratio (which are risk factors for cardiovascular disease). The same authors have shown positive effects of broccoli sprout intake on insulin resistance in type 2 diabetic patients [19,20].

The consumption of strong- and bitter-tasting vegetables from traditional cultivars has been demonstrated to improve the health status of type 2 diabetics in a 3-month clinical intervention. [21]. The group receiving a high daily intake of strong- and bitter-tasting root vegetable and cabbage cultivars demonstrated significant health improvements compared to the group receiving sweet- and mild-tasting vegetables. Although some beneficial effects were detected for consumption of mild, sweet-tasting vegetables, the strong and bitter vegetables had the greatest beneficial impact on insulin sensitivity, body fat mass, and blood pressure, as well as improved glycemic control [21].

Brassicas, commonly known as cruciferous vegetables, are a large group of primarily herbaceous plants that include a number of the world’s most commonly cultivated vegetables [6]. Due to their prevalent adoption in the human diet, plants from the Brassica group provide a significant dietary source of nutrients and bioactive compounds [22]. In nature, the Brassica genus represents 37 species, of which 6 are very commonly used in the human diet (*B. nigra* L., *B. oleracea* L., *B. rapa* L., *B. carinata A*., B. *juncea* (L.), and *B. napus* L.) [23].

Brassica oleracea includes a multitude of recognizable commonly cultivated edible vegetables and vegetable parts, such as leaves and stems in kales, inflorescences in cauliflower and broccoli, leaves surrounding the terminal bud in cabbages, enlarged axillary buds in Brussels sprouts, swollen stems in kohlrabi, etc.

Kale (Brassica oleracea var. acephala) is a cruciferous vegetable which is characterized by leaves along the stem. Different kale cultivars have great genetic variability contributing to wide-ranging kale populations across world and a long history of extensive horticultural use. Kale holds particular importance in the culinary culture and dietary habits of populations in Europe, Asia, and the Americas [24].

Nutritional composition studies of kale cultivars report higher content of protein compared to other green leafy vegetables [5]. In kale, nitrogen compounds (predominantly amino acids) constitute about one-third of the dry matter. According to Lisiewska et. al. (2008), in fresh kale leaves with midribs removed, the dominant amino acids were glutamic acid (12% of total amino acid content), proline (12% of total amino acid content), and aspartic acid (10% of total amino acid content). Furthermore, the proportions of leucine, lysine, valine, arginine, and alanine were in the range of 6–8%, and tyrosine, phenylalanine, threonine, histidine, serine, and glycine varied from 3–5%. Sulphur-containing amino acids were present in the lowest amounts (cystine (1.6%) and methionine (2%)). The authors report that kale leaves, both fresh and processed, can be a good source of proteins and amino acids. The proportion of essential amino acids was 43–46% of total amino acid content, with the lowest concentrations being lysine and leucine [25].

Kale is also particularly rich in vitamins, minerals, dietary fiber, and antioxidative compounds [25]. Kale is an excellent source of vitamin C, pro-vitamin A, lutein, and glucosinolates. According to Becerra-Moreno et. al. (2013), one serving size of kale provides more than 100% of the recommended daily intake of vitamin A and more than 40% of the recommended daily intake of vitamin C [26]. The mineral composition in kale includes sodium (4.69 mg/100 g), potassium (7.03 mg/100 g), calcium (4.05 mg/100 g), iron (8.94 mg/100 g), zinc (2.16 mg/100 g), and magnesium (6.69 mg/100 g). Hence, kale can be considered as a good source of Ca, K, Fe, Na, and Mg and fair source of Zn [27]. Available data confirms that kale is a whole food, low in calories, which can provide significant quantities of daily essential minerals and prebiotic carbohydrates [28].

Besides its promising nutrient content, kale contains high levels of secondary bioactive plant metabolites (phytochemicals) that can exert beneficial health effects in humans. A large body of the research into the health benefits of kale focusses on the content of glucosinolates and phenolic compounds. Phenolic compounds have been extensively investigated for their effects in the management of obesity, type 2 diabetes, metabolic syndrome, neurodegenerative diseases, atherosclerosis, and cancer. It was reported that phenolic compounds, in synergy with other compounds, significantly contribute to the biological activity of Brassica vegetables. Their level in kale varies depending on growth stage, variety, environmental conditions, and geographical location [24].

Glucosinolates are the main class of secondary plant metabolites found in cruciferous vegetables [7]. Glucosinolates contain nitrogen and sulfur within a chemical structure consisting of a β-D-glucopyranose residue linked via a sulfur atom to a (Z)-N-hydroximinosulfate ester, with a variable side chain derived from amino acids [29]. Hwang et. al. (2019) identified various glucosinolates in fresh kale, including glucoraphanin, sinigrin, gluconapin, gluconasturtiin, glucoerucin, glucobrasscin, and 4-methoxylglucobrassicin [30]. The glucosinolate content of kale is dependent upon environmental factors, phenological stage of plant growth, plant parts, and level of insect damage. Different health-promoting properties have been associated with glucosinolates and their hydrolysis products such as isothiocyanates, thiocyanates, nitriles, and epithionitriles [28]. Glucosinolates, and particularly their metabolite isothiocyanates, possess bactericidal, fungicidal, anti-cancer, and anti-inflammatory properties. Furthermore, some potential effects on metabolic parameters such as lipid and glucose metabolism have been reported [31,32,33].

Findings from Kondo et. al. (2016) suggest that the intake of kale suppresses postprandial increases in plasma glucose levels [34]. According to Chung et. al. (2012), the supplementation of regular meals with kale juice may favorably affect serum lipid profiles and could lower the risks of coronary artery disease in adult men with hypercholesterolemia [35].

In light of the current evidence supporting the beneficial effects of kale, the present study aimed to investigate the effects of freeze-dried kale supplementation on various metabolic biomarkers in type 2 diabetic patients. To simplify intake, the freeze-dried kale intervention was incorporated into a convenient bar and administered to T2D patients daily for 3 months and compared to placebo.

## 2. Materials and Methods

This study was carried out according to the guidelines in the Declaration of Helsinki and approved by The Danish Ethics Committee (1-10-72-294-19), the Danish Data Protection Agency, and the study protocol was registered publicly on ClinicalTrials.gov (NCT04298970).

This study was conducted at Aarhus University Hospital, 8200 Aarhus N, Denmark, at the Department of Diabetes and Hormone Diseases. It was designed as a double-blind, randomized, placebo-controlled clinical trial and conducted with type 2 diabetes patients over the period of 3 months.

In total, 4 visits were required from all participants. Some participants were requested to attend an additional visit to the research unit due to lack of availability of the DEXA scanners (making a total of 5 visits for those participants).

Potentially eligible patients were informed about the study through brochures with a detailed project description and also provided a verbal explanation by the research staff. After being given time to consider, consent was obtained from patients that agreed to participate, and they were enrolled into the study. After enrolment but prior to the first study visit, participants arranged personal visit times with the research team and received 3-day dietary questionnaires that were sent to each participant by mail. Baseline information and measurements such as gender, age, medication intake, height, weight, waist circumference, blood pressure, and non-fasting blood glucose measurements were taken. Each participant was randomly assigned either to the green kale group or placebo group (X or Y group) by computer-generated code. Moreover, the participants were asked to complete a 24 h blood pressure measurement with an ambulatory monitor. All participants were instructed not to change their eating, exercise habits, smoking habits, or their medication for the duration of the trial.

On visit day 2 (the next day), participants arrived after a period of overnight fasting to conduct an oral glucose tolerance test (OGTT) and for blood sampling for the biochemical analysis. The OGTT was performed using s 75 g glucose solution. The blood samples were taken at 9 timepoints: −15, 0, 15, 30, 45, 60, 120, 180, and 240 min following ingestion of the glucose solution (within 15 min). Furthermore, urine and fecal samples, together with 24 h blood pressure monitor data, were collected and full-body DEXA scanning was performed. After completing all examinations, the participants were given their respective frozen bars (placebo or kale) and provided with further study instructions. Participants were required to consume 3–4 bars per day and recommended to consume the bars prior to a meal. Participants were also directed to collect the bar wrappings and return them to the research unit to be registered in order to evaluate compliance.

At the end of the three-month period, participants repeated the two-day visit, and all the sampling and measurements were repeated again for comparison. The participants were asked again to fill out 3-day diet questionnaires and bring empty bar wrappers to be counted for compliance. The study timeline is shown in Figure 1.

### 2.1. Recruitment

Participants were recruited by responding to a newspaper announcement, official study flyers, and via social networks. Participants were recruited from the diabetic Danish population from July 2020 to October 2020. Initially, patients were screened for eligibility by telephone, and if they met the inclusion criteria, they were invited to the hospital for an interview and a screening visit.

The inclusion criteria for this trial required participants to be between 30 and 75 years old, have type 2 diabetes (T2D) managed through dietary changes and/or oral anti-diabetic medications, have been diagnosed with T2D after the age of 30, have a body mass index (BMI) between 23 and 40 kg/m^2^;, and have a fasting blood glucose level greater than 4 mM and less than 12 mM, as well as an HbA1c level greater than 43 mmol/mol and less than or equal to 108 mmol/mol. The exclusion criteria were as follows: if participants were currently part of another clinical trial or participated in one recently (≤the last 6 months); if they were being treated with insulin, GLP-1, or other intravenous diabetes medications; and if they had atherosclerosis, psychiatric conditions, nerve and/or kidney disease, acute illness, alcohol or drug abuse, blood pressure ≥ 160/110 mmHg, or pregnancy or breastfeeding. Eligibility for this study was assessed by the research team based on the screening visit.

A power calculation was performed for the primary outcome, insulin sensitivity, as measured by iAUC glucose 240 min from the OGTT. At 80% power and 5% significance, the minimum number of participants required to allow the detection of a difference of 10% in change of AUC glucose 240 min after 12 weeks treatment was estimated to be 28. Adjusting for an anticipated drop-out rate of 20% this gave a total number of patients required to be enrolled of 34.

### 2.2. Kale Bar and Placebo Bar Formulation

Both the green kale and placebo bars were produced by NATURLI’ Foods A/S, Vejen, Denmark. The freeze-dried kale powder used in the intervention bar was provided by Green Gourmet, Randers, Denmark. The powder was 100% pure Brassica oleracea var. Acephala convar Sabellica, also known as Danish Curley Kale. A total of 1g of the intervention powder corresponded to 12 g of fresh kale. The nutritional and energy properties of the powder are shown in detail in Table 1.

Kale was only present in the intervention bar and not the placebo. Apart from this, the same ingredients were used in both bars, with the exception of freeze-dried kale in the intervention bar and cornmeal in the control bar. The ingredients of the bars are shown in Table 2 and Table 3. The energy levels and macronutrient proportions of the placebo and kale bars were analyzed using Vitakost.dk, a professional diet calculator [36], and are presented in Table 4.

### 2.3. Biochemical Analysis

Biochemical analyses were carried out by the Medical Biochemistry Department and Biochemistry Department of Aarhus University Hospital, Denmark. The laboratory analysis was blinded, and all procedures were carried out in accordance with the manufacturer’s instructions.

Fasted blood samples of all participants were collected at the beginning of this study (0 w) and after 12 weeks using either EDTA and lithium heparin-evacuated tubes and temporarily stored on ice. The samples were then centrifuged at 4 °C at 3500× *g* within 30 min. Plasma was then separated into aliquots and stored at −80 °C. The samples from each subject were then analyzed in one batch to reduce the inter-assay variation after the completion of this study.

Plasma glucose was assessed using an enzymatic reference method (Roche Diagnostics GmbH, Mannheim, Germany) on the Cobas C111 system. Fasted plasma lipid profile including triglycerides, total cholesterol, and HDL and LDL cholesterol was measured using enzymatic colorimetric assays on the Cobas C111 system (Roche Diagnostics GmbH, Mannheim, Germany). Glycated hemoglobin HbA1c was analyzed using HPLC on an HLC-723 GHb G7 (Tosoh Europe N.V., Tessenderlo, Belgium).

Plasma insulin was measured with an enzyme-linked immunosorbent assay (K6219, Dako, Glostrup, Denmark), and total GLP-1 concentration in plasma was measured using a commercial total GLP-1 ELISA kit (EMD Milipore, Billerica, MA, USA) according to the manufacturer’s recommendation.

Based on fasting plasma glucose and insulin concentrations, the homeostatic model assessment (HOMA) assesses insulin resistance (IR) by the HOMA-IR.

Body composition was measured using a whole-body dual-energy X-ray absorptiometry (DEXA) scan at weeks 0 and 12. All scans were performed by a trained radiographer using a Norland XR-800 dual-energy X-ray absorptiometry scanner (Norland Cooper, Surgical, Trumbull, CT, USA).

### 2.4. Data Analysis

Analysis included all participants who completed this study and complied with the instructions of this study. The analysis was blinded for the researchers.

The data were analyzed, and the graphs were created using GraphPad Prism 4.3 (GraphPad Software Inc., San Diego, CA, USA).

All datasets were assessed for normality prior to further statistical testing via visualization of the data (QQ plots and histograms) and by the D’Agostino–Pearson test. Baseline comparison was assessed by the unpaired Student’s *t*-test. Absolute change from baseline was calculated after the 12-week study period and assessed by unpaired Student’s *t*-test. The differences within the groups were assessed by the paired Student’s *t*-test. All the data are presented as a mean value ± standard error of the mean (SEM). *p* ≤ 0.05 was considered significant. The primary endpoint was change in the area under the curve (AUC) values of glucose concentrations over the period of 240 min, determined in the OGTT. 

## 3. Results

Thirty participants completed the 12 weeks of intervention, with thirteen participants dropping out (Figure 2). The intervention was implemented as intended, and there were no adverse events associated with the intervention. Participants were well-matched at the time of randomization (week 0), with no statistical differences in any baseline values between the two groups. Table 5 shows the mean and range of baseline characteristics of the subjects.

### 3.1. Glucose Regulation

Fasting blood glucose measurements using OneTouch measurements were taken at the screening visits (week 0) and at study completion (week 12). The results demonstrated an increase of 0.93 ± 0.40 mmol/L within the control group compared to a decrease in the intervention group of −0.5 ± 0.76 mM/L. The change between the groups was not significant (*p* = 0.19, Figure 3).

The change in fasted HbA1c levels from the start (week 0) to end (week 12) of this study for the control and the intervention group was assessed. A decrease of −1.368 ± 0.8591 was noted in the HbA1c of the intervention group from the start to end of this study compared to control, where HbA1c was shown to increase by 1.364 ± 0.7894. The decrease in HbA1c in the group treated with the kale bars was significant compared to control (*p* = 0.04, Figure 4).

Change in AUC of glucose concentrations over the period of 240 min from the start (week 0) to end (week 12) for the control and the intervention group was assessed. An increase in the AUC of glucose was seen within the control group from the start to end of this study at 56.40 ± 70 mM but was not significant (*p* = 0.444), whereas a significant decrease was found within the intervention group −199.9 ± 87 mM (*p* = 0.035). A notable trend was observed between the groups after the 12-week intervention period when comparing the change in area under the curve (ΔAUC) for glucose (*p* = 0.058); however, this result did not reach statistical significance. (Figure 5A).

Results from the assessment of insulin resistance demonstrated a significant difference in the change in HOMA-IR between the intervention group and the control group (Figure 5B). The HOMA-IR levels in the intervention group showed a significant decrease from the start to the end of this study (−0.95 ± 0.34) compared to the control group (0.15 ± 0.28) (*p* < 0.03, Figure 5B).

GLP-1 was measured in both groups. After the 12-week intervention, a decrease was seen in the control group −0.58 ± 1.71 pM and an increase in the intervention group 0.41 ± 1.47 pM; however, there was no significant difference seen between the groups.

### 3.2. Body Weight and Composition

Body weight was measured at the beginning of this study and again after 12 weeks. A significant reduction in body weight (Figure 6) was observed in the intervention group compared to control (*p* = 0.037). The reduction of body weight within the intervention group was −0.95 ± 0.35 kg (*p* = 0.012) compared to no change within the control group (0.15 ± 0.29 kg).

The waist circumference data obtained demonstrated a significant decrease of −5.8 ± 0.9 cm (*p* = 0.0001) within the intervention group and a significant reduction within the control group of −33 cm (*p* = 0.02). There was no significant difference between the groups (*p* = 0.11). The waist circumference results are presented in Figure 7.

DEXA scans were utilized to determine the change in total body fat mass in % of both groups. The control (placebo) group increased their body fat percentage from week 0 to week 12 by 0.9%, while the intervention group decrease their body fat by −4.7%. However, there was no significant difference between the two groups (*p* = 0.88) or within groups.

### 3.3. Plasma Lipid Levels

The change of triglycerides (Figure 8A) of blood samples was −0.09 ± 0.11 mmol/mL in the control group and 0.7 ± 0.1 mmol/mL for the intervention group. No significant difference was observed between the groups. The results for total levels of cholesterol (TC) are presented in Figure 8B. Changes in TC were −0.05 ± 0.21 mmol/mL for the control and 0.074 ± 0.08 mmol/mL for the intervention group. Figure 8C presents the change in the LDL levels over the 12-week study period. There was a tendency for LDL to be increased in the control group compared to the intervention group; however, this was not significant.

There was an increase within the control group in the levels of HDL cholesterol (Figure 8D) of 0.100 ± 0.04 mmol/mL (*p* = 0.024) and no significant change within the intervention group of 0.05 ± 0.03 mmol/mL. The difference observed between the groups was not significant.

### 3.4. Blood Pressure

The results of 24 h blood pressure measurements are presented in Table 6. The data are presented as mean ± SEM. No significant difference was observed between or within the groups from the baseline to the end of 12-week study period.

### 3.5. Caloric Intake

The changes in caloric intake in the intervention and control groups are presented in Figure 9. The change in calorie intake for the control group was 248.5 ± 156.4 kcal and −62.75 ± 68.57 kcal for the intervention group. The difference between the groups was significant (*p* = 0.04).

## 4. Discussion

This randomized, controlled, parallel intervention study was designed to investigate, for the first time, if high dietary intake of kale (Brassica oleracea L. var. acephala) in the form of freeze-dried kale powder incorporated into a bar can significantly improve key disease characteristics of T2D patients compared to a control group. This 12-week intervention study found health improvements in the kale intervention group; namely a significant reduction in HbA1c, insulin resistance (HOMA_IR), body weight, and calorie intake compared to the control. Trends were also shown toward a reduction in fasting blood glucose and LDL-cholesterol for the kale group compared to control [21].

The OGTT tests and fasting plasma glucose levels illustrate a clear tendency for a reduction of fasting glucose in the intervention kale group, while a small increase was seen in the control group (Figure 5A) (*p* = 0.058). A significant decrease in HOMA-IR and HbA1c was observed in the intervention group compared to increases in both parameters for the control (Figure 5B (*p* = 0.03) and Figure 4 (*p* = 0.04), respectively). HOMA-IR is one of the pivotal indices of tissue insulin resistance. HOMA-IR (Figure 5B) was significantly improved in the T2D group after the intervention, which clearly indicates an improvement in insulin sensitivity. These findings are in agreement with other findings; however, the results should be interpreted carefully. A study by Thorup et al. (2021) illustrates a considerable positive effect of high daily intake of root vegetables and, in particular, cabbages, on glucose control and insulin sensitivity in T2D patients in the short term (3-months) [21], where HOMA-IR was also significantly reduced in the intervention group compared to a control group [21].

The reduction of HbA1c could indicate a long-term improvement in glycemic control of diabetic patients due to high kale consumption. The limited literature available on the effects of kale on glycemic control suggest that beneficial effects are, to an extent, driven by certain genetic and epigenetic factors. A human intervention study by Han et al. (2015) suggests that a decrease of fasting blood glucose after long-term kale supplementation may be dependent on glutathione S-transferase genetic polymorphisms of individuals. After a 6-week kale juice intervention, a significant reduction in fasting blood glucose levels was observed only in the group with a specific genotype [37]. Another randomized controlled trial involving humans found similar significant benefits on HbA1c levels. In this study, a group of patients with T2D followed different diets, including a daily intake of 500 grams of vegetables, such as root vegetables and cabbages, for 12 weeks, compared to a control group. Another human randomized controlled trial showed similar significant beneficial effects on HbA1c, where a group of T2D patients were allocated different diets (including 500 g daily intake of vegetables containing root vegetables and cabbages) for 12 weeks compared to a control group (who only consumed 120 grams of vegetables per day) [21].

Based on the WHO standards for BMI for adults over 20 years old, the subjects from both groups in the present study are considered to be in a pre-obesity state (with BMI measurements at the baseline close to 30 kg/m2) [38]. The BW was statistically decreased in the freeze-dried kale bar group compared to no change within the control group. Weight loss in overweight and obese individuals with T2D has been associated with significant improvements in cardiovascular disease risk factors and substantial reductions in mortality [39,40]. Even though this intervention was not primarily designed for weight loss, the observed positive effects of kale on body weight could consequently help improve diabetes status and its complications in the long term, likely having pronounced effects in obese patients. [41,42,43]. We fully acknowledge that the average weight loss for the kale group did not achieve clinical significance; as the inclusion criteria included both normal and overweight patients, the effects are likely due to large individual differences. A further clinical trial including only obese or overweight patients would be better-suited to elucidate the effects of kale bars on weight loss. Following weight loss, a reduction was seen within both groups in waist circumference; however, no significant difference was detected between the groups. This may indicate potential beneficial effects of freeze-dried kale on body fat distribution given a longer-term treatment. Abdominal obesity assessed by anthropometry may indicate a reduction in excess visceral fat, which is a risk factor for various negative cardiovascular outcomes [44]. The maintenance of healthy weight and waist circumference are highly important for both healthy individuals and individuals living with diabetes [45]. The potential of this kale intervention to promote healthy weight loss is supported by the present study, with significant weight loss in the intervention group compared to control. Weight loss was largely driven by reductions in caloric intake. The kale group attained a greater level of dietary reduction in calories than the placebo group after 12 weeks (Figure 9, *p* = 0.04). [46]. Vegetables are rich sources of dietary fibers, and research has shown that food with a high fiber content causes early satiation, enhanced sensations of satiety, decreased subsequent hunger, a decrease in insulin response following meals, and the slowing of gastric emptying, thus increasing macronutrient absorption [47]. The kale bars in this study provided a high amount of dietary fibers (25.8 g/day) [47]. Furthermore, research has shown that the water extract of freeze-dried kale powder can stimulate GLP-1 secretion in vitro [48]. GLP-1, known as a ‘satiety hormone’, has been reported to enhance insulin synthesis and secretion, delay gastric emptying and gastrointestinal motility, and reduce food intake [49]. In our study, we observed no significant differences between the groups, possibly due to substantial interindividual variability among the patients in their ability to synthesize GLP-1 and the rapid elimination of GLP-1 from the body [50]. Since the samples were taken after fasting, baseline GLP-1 levels were likely low among all participants, possibly masking intervention effects as GLP-1 is typically released after meals. However, it is noteworthy that GLP-1 levels did increase in the intervention group and decreased in the control group. This suggests that a high intake of kale may have a modest impact on GLP-1 levels, potentially influencing patients’ food intake and, consequently, their caloric intake and body weight. Nevertheless, this hypothesis requires further investigation.

It is plausible that, implementing freeze-dried kale bars into the regular diet may ameliorate satiety, decreasing overall daily caloric intake, consequently promoting weight loss. In this study, DEXA scans facilitated a more in-depth investigation of body composition. DEXA is a precise, accurate, non-invasive, safe, and convenient technique founded on a three-compartment model separating the body into total body mineral mass, fat mass, and lean tissue mass, the latter being the remaining bone-free, fat-free tissue mass [51,52]. The control group increased their body fat % from week 0 to week 12 by 0.9%, while the intervention group decrease their body fat by −4.7%. Although there was no significant difference between the two groups, losses in weight of 5%–15% are considered clinically meaningful in obese patients (affecting diabetes remission and cholesterol) and may reduce depression, joint pain, and sexual function [53].

The clinical conditions commonly associated with T2D are hypertension and dyslipidemia, which are critical risk factors for CVD. CVD is the leading cause of morbidity and mortality in individuals affected by T2DM. The rates of CVD mortality are two to four times higher in diabetic patients compared with the non-diabetic population [54,55]. It is highly important for T2D patients to have BP and lipid levels within the recommended levels [56]. Participants in this study had mean baseline levels of total cholesterol, LDL and HDL cholesterol, and triglycerides that were approximately within the recommended levels (Table 5). The lipid profiles were not significantly affected by the intervention (Figure 8). It should be noted that this research was conducted in the second half of the year, accompanied by the arrival of colder weather. Studies have shown lipid concentrations vary across the seasons with peak values typically occurring in the winter. Lipid levels are generally higher in the colder periods due to reduced activity and exercise levels and increased intake of “heavy” (higher-fat-containing) foods [57]. This seasonal total cholesterol increase followed both groups in this study. However, it should be noted that an elevation in LDL cholesterol levels was observed only in the control group, while the LDL levels remained unchanged from the baseline to 12 weeks in the intervention group. This suggests that the kale bars used in the intervention group could help in plasma LDL cholesterol regulation. The effects of kale in this study on cholesterol are in supported by prior research [58]. Kale is a complex food matrix suggested to possess a high potential to bind bile acids. If the food fractions bind bile acids, their reabsorption is then prevented and the conversion of plasma and liver cholesterol to additional bile acids is stimulated [59]. It should be noted that the participants in both groups were already very well-managed in terms of blood lipids from the beginning.

In this trial, ambulatory BP monitoring with repeated measurements over 24 h was used. This method is less variable and more accurately detects changes in BP compared to clinical measurements [60]. BP levels are shown to be affected by genetic and behavioral factors as well as psychosocial factors [61]. Although systolic and diastolic blood pressure depend on each other, systolic BP varies to a greater degree than diastolic BP [62]. Furthermore, night-time systolic BP predicts outcomes more accurately than daytime BP [60]. There was a trend toward a greater decrease in night-time diastolic blood pressure in the intervention group compared to control, which could indicate positive effects of kale on blood pressure (Table 6). Although the reduction was not statistically significant, it could be of clinical importance considering that even a small (1–4 mmHg) long-term reduction in BP is estimated to reduce the risk of cardiovascular mortality [63]. The present study detected a decrease in the kale intervention group of about −3.9 to −3.3 mmHg; Table 6.

A total of 8.75 g of freeze-dried kale powder was incorporated into each bar used in the intervention group. This amount corresponds to 114 g of fresh kale. Therefore, the consumption of three bars a day would increase vegetable intake to levels corresponding to approx. 341g of fresh kale per day. The WHO recommends a minimum of 400 g of fruit and vegetables per day, excluding starchy roots [64]. Hence, the consumption of bars provided a recommended daily vegetable intake in the intervention group. The practicality of this type of vegetable intake should be emphasized as an advantage of the kale bar product. It does not require any preparation, except defrosting a few minutes before consumption. However, some ingredients could be reconsidered in order to improve the functionality of the bar. The bar features rice-based products such as rice drink and rice porridge. The glycemic index of rice is higher than that of any other starch-containing food. High-GI foods can quickly raise blood glucose to high levels and affect insulin resistance, which may adversely affect the maintenance of glycemic homeostasis [65]. Furthermore, the analysis showed that kale powder contains active isothiocyanates and their precursors in desirable concentrations; however, the bar containing powder was not analyzed after processing. Although the bars were frozen immediately after processing, it is possible that ITCs were affected at some point prior to consumption. The literature indicates low ITC stability in various media. According to Luang-In & Rossiter (2015), a decrease in concentrations of five different ITCs tested, from 1.0 mM to 0.2–0.4 mM, occur in aqueous media, within 8 h [66]. Thereby, caution should be taken when interpreting the results regarding ITCs due to their instability in media and buffers.

A limitation of this research is the small sample size. The power calculation required 28 participants to complete this study. Of the 43 participants enrolled in this study, only 30 finished according to the protocol. This likely significantly affected the end results of this study. Since this research was conducted in 2020–2021 during a worldwide Covid-19 pandemic, participant recruitment and the conduct of the clinical trial were severely hindered. In addition to the discomfort participants were experiencing in keeping social distance, wearing masks, and adhering to excessive hygienic measures, the fear for their health played a major role in both the recruitment and retention of participants. A total of 25 participants were enrolled in the intervention group and 18 in the control. A higher dropout rate occurred in the control group, leaving this group at only 11 participants at the end of this study, compared to 19 in the intervention group. Another limitation of this study lies in the self-reported food questionnaires which are subject to bias. The strengths of this study lie within its design. The trial was randomized, placebo-controlled, and double-blinded. Generally, RCTs are characterized by the highest reliability [67]. In order to observe clearer effects of freeze-dried kale powder on health, a similar study with a larger sample of participants and a longer duration is recommended in the future.

## 5. Conclusions

The health effects of a daily intake of 26.25 g of freeze-dried kale powder (corresponding to approx. 351 g of fresh kale) incorporated into a bar on T2D patients were investigated in this randomized, double-blinded, placebo-controlled trial conducted over a period of 12 weeks in 30 diabetic patients. The kale intervention showed significant positive effects on HbA1c, HOMA-IR, body weight, and caloric intake compared to control. Some positive trends were observed in fasting blood glucose OGTT, blood pressure, and LDL-cholesterol levels. Despite limitations, the findings in this study support that the daily intake of freeze-dried kale powder can exert beneficial health effects on key factors in T2D patients (glycemic control, weight loss, caloric intake). Freeze-dried kale powder may also improve feelings of satiety, blood pressure, and lipid metabolism. Additional research with a larger sample size is recommended to achieve more robust insights into the positive health effects of kale bars. Further research has the potential to contribute to new dietary recommendations for T2D patients.

## Figures and Tables

**Figure 1 nutrients-16-03641-f001:**
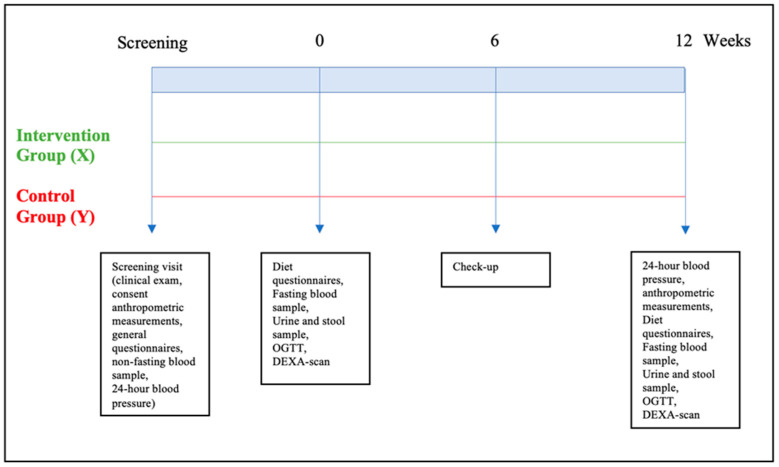
Study timeline.

**Figure 2 nutrients-16-03641-f002:**
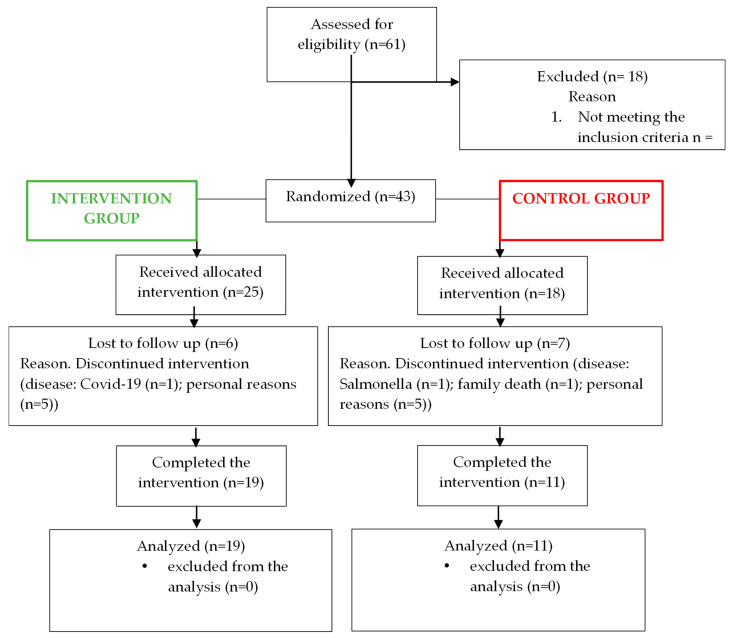
Study flow diagram.

**Figure 3 nutrients-16-03641-f003:**
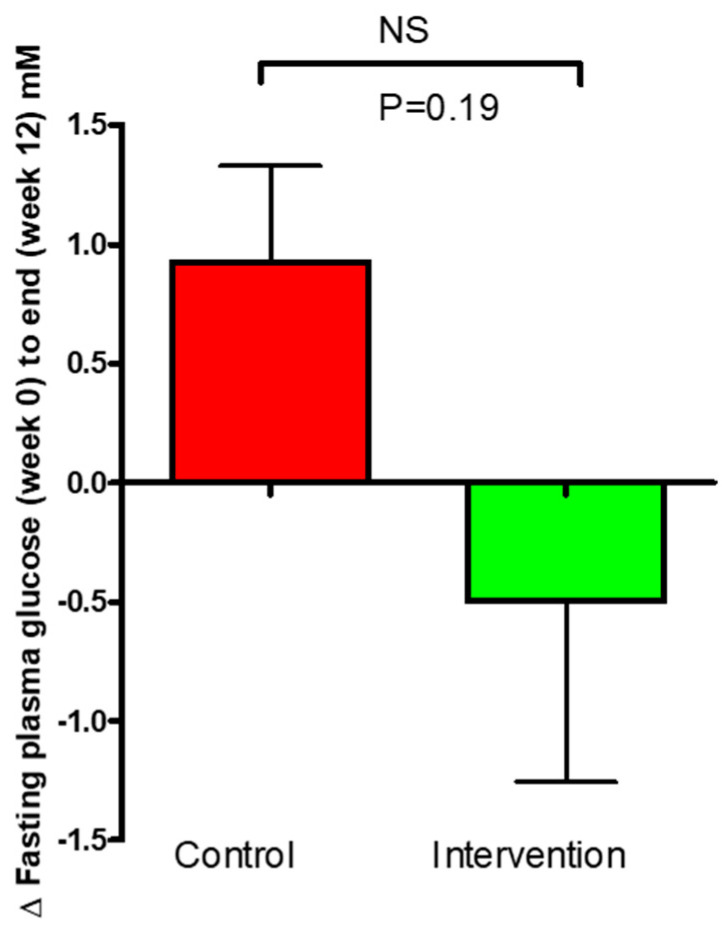
Change (Δ) in plasma glucose OneTouch for the control and intervention groups over the 12- week study period. The data are presented as mean ± SEM.

**Figure 4 nutrients-16-03641-f004:**
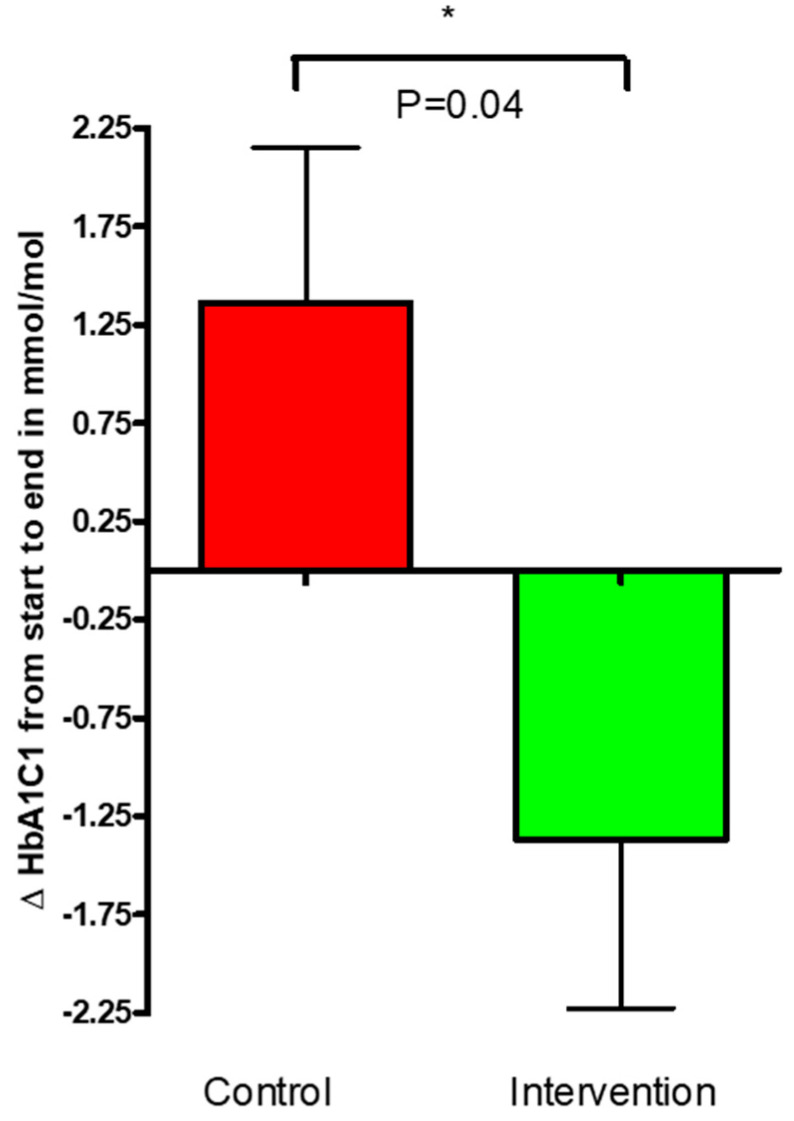
Change (Δ) in hemoglobin A1c for the control and intervention groups over the 12-week study period. The data are presented as mean ± SEM. * *p* < 0.05.

**Figure 5 nutrients-16-03641-f005:**
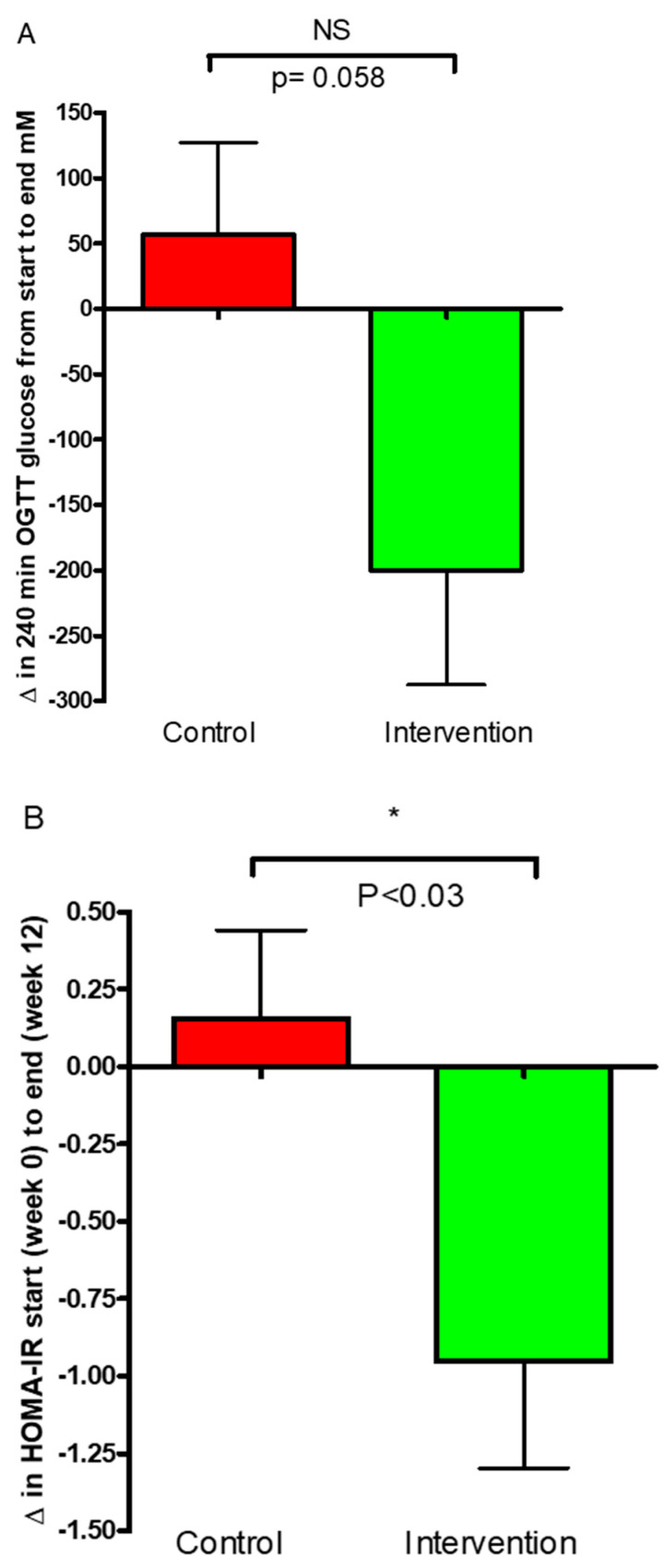
(**A**,**B**). 5A: Results of oral glucose tolerance test. Change in total area under the curve (AUC) values for the control and intervention groups at the start (week 0) and end (week 12) of this study are shown. 5B: HOMA-IR levels in the intervention group showed a significant decrease from the start to the end of this study (−0.95 ± 0.34) compared to the control group (0.15 ± 0.28) The data are presented as mean ± SEM. * *p* < 0.05; ns: not significant.

**Figure 6 nutrients-16-03641-f006:**
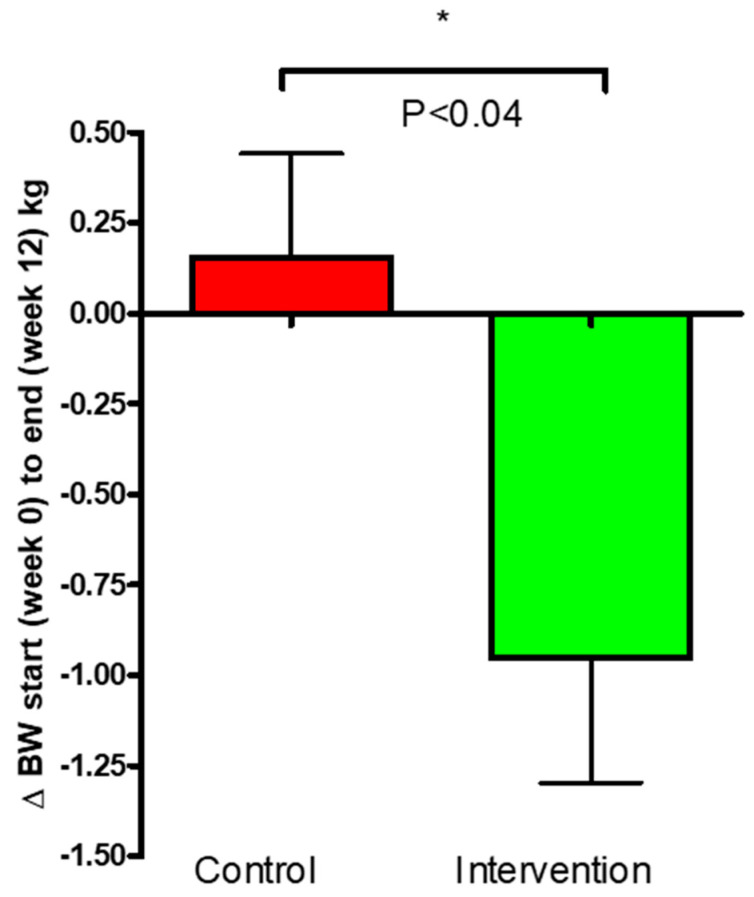
Changes (Δ) in body weight for the control and intervention groups over the 12-week study period. The data are presented as mean ± SEM. (*) *p* ≤ 0.05.

**Figure 7 nutrients-16-03641-f007:**
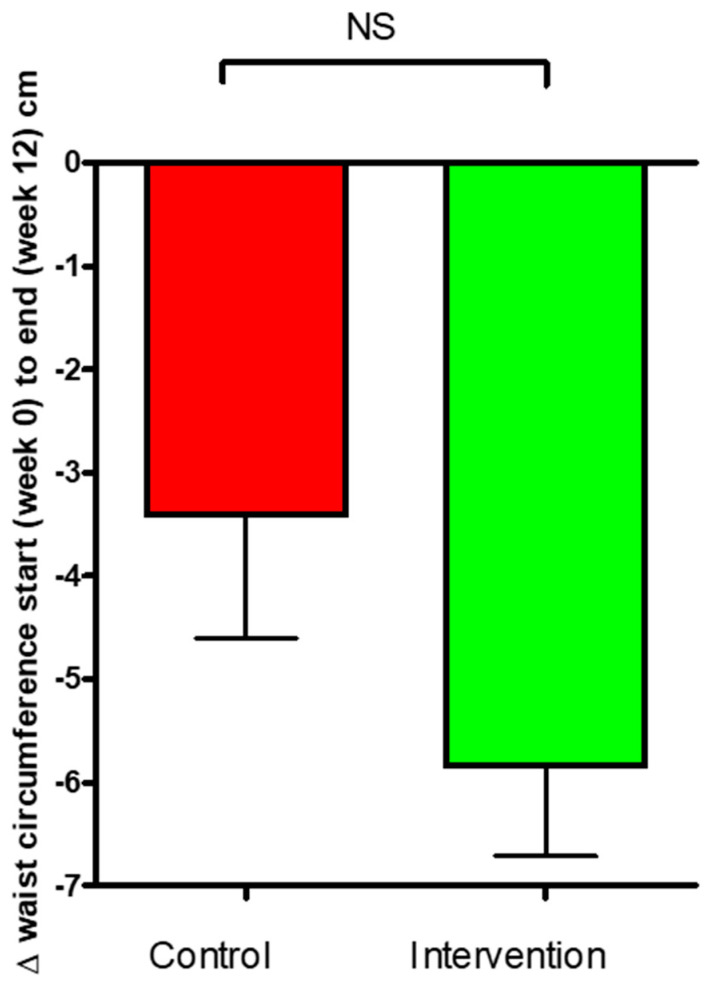
Change (Δ) in waist circumference for the control and intervention groups over the 12-week study period. The data are presented as mean ± SEM; NS, not significant.

**Figure 8 nutrients-16-03641-f008:**
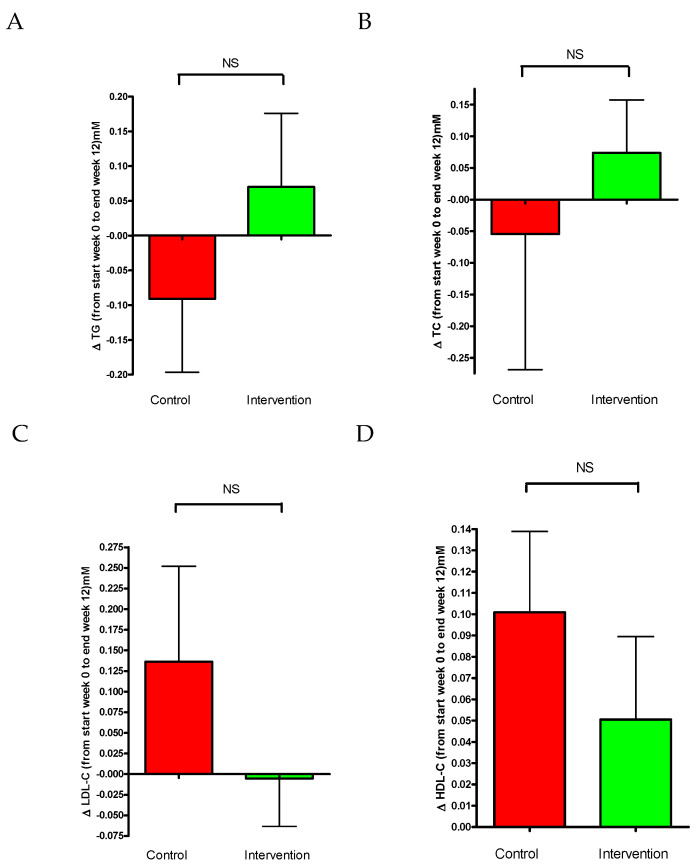
Change (Δ) in plasma lipid ((**A**) Tg, (**B**) TC, (**C**) LDL-C, (**D**) HDL-C) levels for the control and intervention groups over 12 weeks. The data are presented as mean ± SEM; NS, not significant.

**Figure 9 nutrients-16-03641-f009:**
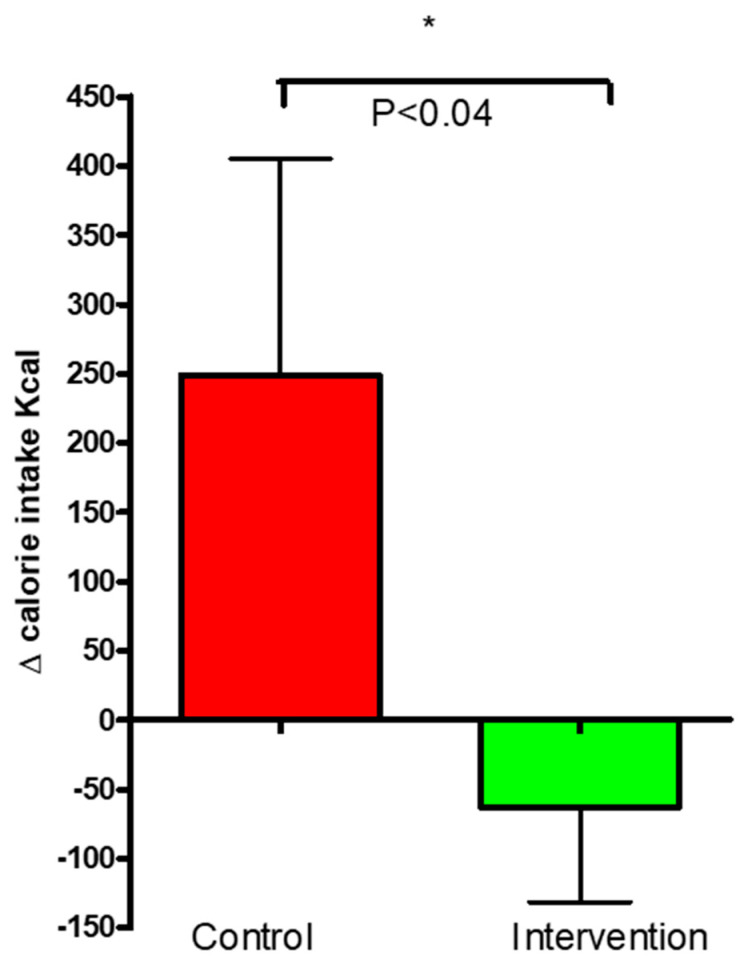
Change (Δ) in caloric intake for the control and intervention groups over 12 weeks. The data are presented as mean ± SEM. * *p* ≤ 0.05.

**Table 1 nutrients-16-03641-t001:** Nutrient and energy content of freeze-dried kale powder per 100 g.

Energy/100 g	312 kcal/1295 kJ
Total fat	7.3 g
Saturated fat	0.8 g
Monounsaturated fat	0.5 g
Polyunsaturated fat	2.0 g
Carbohydrates	16 g
Sugars	5.5 g
Fiber	41 g
Proteins	25 g
Amino acids	Valine—1.27 g Isoleucine—0.939 g Leucine—0.139 g Phenylalanine—1.15 g Lysine—0.205 g Histidine—0.145 g Tryptophan—0.386 g Methionine—0.416 g Threonine—1.07 g
NaCl	0.08 g
Minerals & Vitamins	Potassium—2900 mg Calcium—1800 mg Phosphate—480 mg Magnesium—170 mg Vitamin C—350 mg Vitamin K1—330 mg

**Table 2 nutrients-16-03641-t002:** Ingredient composition of kale bar used in the intervention group.

Ingredient	Amount in One Bar	Energy
Natural rice drink, organic, Naturli’	19.85 g	10.75 kcal
Rapeseed oil, organic, Änglamark	4 g	35.35 kcal
Cocoa powder	5 g	20.1 kcal
Artificial sweetener, ISIS	3.5 g	7.45 kcal
Coconut, organic, Änglamark	5 g	31.05 kcal
Green kale powder	8.75 g	27.3 kcal
Rice porridge, Pama	1.5 g	5.4 kcal
TOTAL	47.6 g	137.5 kcal

**Table 3 nutrients-16-03641-t003:** Ingredient composition of placebo bar used in the control group.

Ingredient	Amount in One Bar	Energy
Cornmeal, organic, Urtekram	2 g	7.4 kcal
Natural rice drink, organic, Naturli’	23 g	12.5 kcal
Rapeseed oil, organic, Änglamark	4 g	35.35 kcal
Cocoa powder	5 g	20.1 kcal
Artificial sweetener, ISIS	3.5 g	7.45 kcal
Coconut, organic, Änglamark	5 g	31.05 kcal
Rice porridge, Pama	8.75 g	31.6 kcal
TOTAL	51.5 g	145.5 kcal

**Table 4 nutrients-16-03641-t004:** Energy levels and macronutrient proportions of the kale and the placebo bars.

	Placebo Bar (51.5 g)	Green Kale Bar (47.6 g)
Energy	145.5 kcal	137.5 kcal
Fat	8.5 g	9.05 g
Saturated fat	3.855 g	3.905 g
Monounsaturated fat	2.895 g	2.865 g
Polyunsaturated fat	1.45 g	1.395 g
Carbohydrate	12.65 g	6.45 g
Sugar	2.08 g	2.325 g
Fiber	5.1 g	8.6 g
Protein	2.02 g	3.485 g
NaCl	0.029 g	0.033 g
Alcohol	0 g	0 g

**Table 5 nutrients-16-03641-t005:** Baseline characteristics of participants included in data analysis. The data are presented as mean ± SEM.

	Control Group	Intervention Group	*p*-Value
Participants(n) Men Women	11 5 6	19 17 2	
Age (years)	64.3 ± 2.1	62.7 ± 1.5	0.55
Weight (kg)	89.82 ± 5.46	90.49 ± 3.63	0.88
BMI (kg/m^2^)	29.81 ± 1.52	29.50 ± 1.06	0.87
Waist circumference (cm)	97.95 ± 3.63	99.39 ± 3.21	0.77
Fasting plasma glucose (mM)	7.85 ± 0.26	7.91 ± 0.19	0.84
HbA1c (mM)	48.18 ± 1.8	50.79 ± 2.4	0.39
Triglycerides (mM)	1.46 ± 0.19	1.29 ± 0.10	0.43
Plasma total cholesterol (mM)	4.34 ± 0.5	3.87 ± 0.20	0.40
Plasma HDL-cholesterol (mM)	1.28 ± 0.16	1.22 ± 0.07	0.74
Plasma LDL-cholesterol (mM)	2.55 ± 0.34	2.05 ±0.19	0.22
Systolic blood pressure (mmHg)	126.9 ± 3.27	124.1 ± 3.65	0.57
Diastolic blood pressure (mmHg)	77.73 ± 1.63	74.68 ± 1.33	0.16

**Table 6 nutrients-16-03641-t006:** Changes (Δ) in blood pressure for the control and intervention group over 12 weeks. The data are presented as mean ± SEM.

	z			Intervention		
	Change		P	Change	P	P
Systolic	2.82 ± 2.20		0.23	−0.95 ± 3.42	0.79	0.36
Diastolic	1.27 ± 1.48		0.41	0.74 ± 1.32	0.59	0.79
Systolic/day	3.36 ± 2.53		0.21	0.84 ± 3.22	0.8	0.54
Diastolic/day	1.09 ± 1.73		0.54	1.42 ± 1.23	0.29	0.88
Systolic/night	1.09 ± 2.33		0.65	−3.94 ± 3.27	0.24	0.22
Diastolic/night	−1.36 ± 2.6		0.61	−3.26 ± 1.55	0.05	0.54

## Data Availability

The datasets generated during and/or analyzed in the current study are available from the corresponding author under special circumstances due to privacy or ethical restrictions.

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
