# Peer review of "Beneficial Effects of a Freeze-Dried Kale Bar on Type 2 Diabetes Patients: A Randomized, Double-Blinded, Placebo-Controlled Clinical Trial"

_nutrients, 2024, doi:10.3390/nu16213641_

Round 1

Reviewer 1 Report

Comments and Suggestions for Authors

Due to the number of cases and the poor prognosis for the future, type 2 diabetes is one of the most serious medical problems. Proper control of blood glucose levels is extremely important in its course. This is attempted through pharmacotherapy and a proper diet. Many centers are testing the effects of various dietary supplements on the possibility of better control of sugar levels. One such element is kale. On the other hand, proper determination of glucose levels is extremely important. The most reliable method is currently considered to be the determination of glycosylated hemoglobin levels, which are the result of the reaction of glucose with amino acids of hemoglobin. Since this is a non-enzymatic reaction, glycosylated hemoglobin levels give reliable blood glucose levels in the last 3 months before the test.

The authors preceded the research results presented in the article with an extensive introduction written on the basis of contemporary and well-selected scientific literature.

To obtain their results, the authors used routine tests such as fasting blood glucose, HbA1c  glucose levels, HDL and LDL lipoprotein levels, and blood pressure. They also determined body parameters.

It is worth noting that both the Placebo and Kale bars were prepared in such a way that the impact of the remaining ingredients on the tested parameters was as small as possible, making the obtained results more reliable.

In the analyzed period, most of the parameters studied did not change significantly, as the authors themselves note. It is difficult to agree with the interpretation of Figure 6, which shows changes in the participants' body weight in the analyzed period. It is difficult to consider a change of about 1 kg as significant (line 335), especially if we take into account the determined measurement error. In my opinion, changes in this element should also be considered insignificant.

The changes in the level of hemoglobin HbA1c in the group taking freeze-dried kale powder should be considered valuable and certainly worth noting. Of course, a cautious approach to the obtained results, as the authors are aware, dictates the narrow group of cases studied as well as the time in which the studies were conducted (the Covid-19 epidemic), which significantly changed people's social behavior. However, in my opinion, the results are promising enough to support repeating the studies on a larger scale.

Author Response

Reviewer 1 comments 1: Due to the number of cases and the poor prognosis for the future, type 2 diabetes is one of the most serious medical problems. Proper control of blood glucose levels is extremely important in its course. This is attempted through pharmacotherapy and a proper diet. Many centers are testing the effects of various dietary supplements on the possibility of better control of sugar levels. One such element is kale. On the other hand, proper determination of glucose levels is extremely important. The most reliable method is currently considered to be the determination of glycosylated hemoglobin levels, which are the result of the reaction of glucose with amino acids of hemoglobin. Since this is a non-enzymatic reaction, glycosylated hemoglobin levels give reliable blood glucose levels in the last 3 months before the test.

The authors preceded the research results presented in the article with an extensive introduction written on the basis of contemporary and well-selected scientific literature.

To obtain their results, the authors used routine tests such as fasting blood glucose, HbA1c  glucose levels, HDL and LDL lipoprotein levels, and blood pressure. They also determined body parameters.

It is worth noting that both the Placebo and Kale bars were prepared in such a way that the impact of the remaining ingredients on the tested parameters was as small as possible, making the obtained results more reliable. In the analyzed period, most of the parameters studied did not change significantly, as the authors themselves note.

It is difficult to agree with the interpretation of Figure 6, which shows changes in the participants' body weight in the analyzed period. It is difficult to consider a change of about 1 kg as significant (line 335), especially if we take into account the determined measurement error. In my opinion, changes in this element should also be considered insignificant.

Response 1: Thank you for your point. We agree that an average change of 1 kg is not much, but this is due to individual differences. It is worth noting that some participants lost between 3-5 kg, while others remained unchanged. This, considering that weight loss itself was not a target. We have now added the following text: "We fully acknowledge that the average weight loss for the cabbage group did not achieve clinical significance , as the inclusion criteria included both normal and overweight patients the effects is due to large individual differences. A further clinical trial including only obese or overweight patients would be better suited to elucidate the effects of Kale bars on weight loss" P: 15L424-428

Reviewer 1 comments 2: The changes in the level of hemoglobin HbA1c in the group taking freeze-dried kale powder should be considered valuable and certainly worth noting. Of course, a cautious approach to the obtained results, as the authors are aware, dictates the narrow group of cases studied as well as the time in which the studies were conducted (the Covid-19 epidemic), which significantly changed people's social behavior. However, in my opinion, the results are promising enough to support repeating the studies on a larger scale.

Response 2: Thank you for your point

Reviewer 2 Report

Comments and Suggestions for Authors

In the manuscript entitled “Beneficial effects of freeze-dried kale bar on type 2 diabetes patients: A randomized, double-blinded, placebo controlled clinical trial” the authors in this study the authors examine the effects of daily consumption of freeze-dried kale powder on patients with type 2 diabetes (T2D). The results show a significant reduction in HbA1c, insulin resistance, body weight, and caloric intake in the kale group, with positive trends also in fasting blood glucose and LDL cholesterol.

The work is well written and the experiments are well thought out and well done.

Minor points

  • I suggest that the authors increase the number of participants to improve the statistical power and generalizability of the results, considering the relatively high dropout rate (13 out of 43 initial participants). A study with a larger cohort may provide more robust and conclusive results.
  • In addition, despite the body weight and fat mass measurements, it would be useful to also analyze changes in body fat distribution, such as visceral fat, using DEXA scans to further understand the impact of kale.
  • The authors should also consider including inflammatory biomarkers (e.g. CRP, TNF-α) to assess the effect of kale on inflammation, as it is an important component of type 2 diabetes.
  • The authors should consider exploring the relationship between changes in lipid levels (triglycerides, LDL and HDL cholesterol) and other metabolic markers such as HbA1c or insulin resistance. A correlation analysis could help better understand the impact of kale on cardiovascular risk factors in patients with T2D.

Author Response

Reviewer 2, Comments 1: In the manuscript entitled “Beneficial effects of freeze-dried kale bar on type 2 diabetes patients: A randomized, double-blinded, placebo controlled clinical trial” the authors in this study the authors examine the effects of daily consumption of freeze-dried kale powder on patients with type 2 diabetes (T2D). The results show a significant reduction in HbA1c, insulin resistance, body weight, and caloric intake in the kale group, with positive trends also in fasting blood glucose and LDL cholesterol.

The work is well written and the experiments are well thought out and well done.

Minor points

  • I suggest that the authors increase the number of participants to improve the statistical power and generalizability of the results, considering the relatively high dropout rate (13 out of 43 initial participants). A study with a larger cohort may provide more robust and conclusive results.

Response 1: 

Thank you for your comment. It is always desirable to include as many participants as possible and to have as many as possible complete the study. In our defense, We conducted a power calculation prior to starting the study, based on studies with a similar intervention, and we were able to meet the required number of participants. It is important to remember that this study was carried out under very challenging conditions due to the COVID-19 pandemic, as we also describe in the article, and this likely contributed to the increased dropout rate. Nevertheless, we still feel that we have solid data collection, with the vast majority of participants completing the study, supported by our power calculation.

Reviewer 2 comments 2: 

  • In addition, despite the body weight and fat mass measurements, it would be useful to also analyze changes in body fat distribution, such as visceral fat, using DEXA scans to further understand the impact of kale.

Response 2: Thank you for your point. Since there was no significant difference in body composition in the DEXA scans, we have chosen not to proceed further with this method in this article. However, we plan to write another article where we will include visceral fat in relation to inflammatory biomarkers.

Reviewer 2 comments 3: The authors should also consider including inflammatory biomarkers (e.g. CRP, TNF-α) to assess the effect of kale on inflammation, as it is an important component of type 2 diabetes.

Response 3: Thank you for your advice. As mentioned above, we will aim to write a second article focusing on inflammatory biomarkers in relation to changes in fat deposits, such as visceral fat, which we know influences inflammation.

Reviewer 2 comments 4: 

The authors should consider exploring the relationship between changes in lipid levels (triglycerides, LDL and HDL cholesterol) and other metabolic markers such as HbA1c or insulin resistance. A correlation analysis could help better understand the impact of kale on cardiovascular risk factors in patients with T2D.

Response 4: Thank you for your excellent suggestions. We agree that it could be interesting, but since there is already a considerable amount of data in the article, we are concerned that adding more may reduce clarity. Additionally, as shown in Fig. 8, we did not find significant differences between the groups in relation to lipids. The participants were already very well-managed in terms of blood lipids. We have now added the following to the paper: "It should be noted that the participants in both groups were already very well-managed in terms of blood lipids from the beginning" P 16 L 492-93